# The Impact of COVID-19 on Nursing Homes: Study Design and Population Description

**DOI:** 10.3390/ijerph192416629

**Published:** 2022-12-10

**Authors:** Vicente Rodriguez-Rodriguez, Fermina Rojo-Perez, Carmen Perez de Arenaza Escribano, María-Ángeles Molina-Martínez, Gloria Fernandez-Mayoralas, Diego Sánchez-González, Jose-Manuel Rojo-Abuin, Carmen Rodríguez-Blázquez, Maria João Forjaz, Salomé Martín García

**Affiliations:** 1Grupo de Investigacion sobre Envejecimiento (GIE), IEGD, CSIC, 28037 Madrid, Spain; 2Departamento de Personalidad, Evaluación y Tratamiento Psicológico, Facultad de Psicología, Universidad Nacional de Educación a Distancia (UNED), 28040 Madrid, Spain; 3Departamento de Geografía, Universidad Nacional de Educación a Distancia (UNED), 28040 Madrid, Spain; 4Unidad de Analisis Estadistico (UAE), CCHS, CSIC, 28037 Madrid, Spain; 5National Epidemiology Centre, Carlos III Institute of Health, Network Centre for Biomedical Research in Neurodegenerative Diseases (CIBERNED), 28029 Madrid, Spain; 6National Epidemiology Centre, Carlos III Institute of Health, Health Service Research Network on Chronic Diseases (REDISSEC) and Research Network on Chronicity, Primary Care and Health Promotion (RICAPPS), 28029 Madrid, Spain; 7EULEN Servicios Sociosanitarios, 28023 Madrid, Spain

**Keywords:** nursing homes, COVID-19, physical health, mental health, perception, feelings, coping, residential environment, quality of life, ageism

## Abstract

Nursing homes for the elderly in Spain have experienced high rates of infection and mortality from COVID-19, although rates have varied from one region to another. Madrid is the region where most institutionalized older adults have died from the coronavirus. However, there is little known about the psychosocial and environmental factors involved in the high incidence of COVID-19 among the institutionalised population in this region. This article describes the protocol of a study on nursing homes during the SARS-CoV-2 pandemic in the Autonomous Community of Madrid (hereafter: Region of Madrid or Madrid Region) and provides information on the study design, measures used, and characteristics of the population studied. A questionnaire about life in nursing homes during the COVID-19 pandemic was designed and a total of 447 persons over 60 years of age without cognitive impairment—220 in private nursing homes and 227 in public nursing homes—participated by answering questions about different topics: personal situations during the pandemic, feelings and methods of coping, residential environment, health, quality of life, ageism, and self-perception of ageing. The institutionalised person profile discussed in this study was an old woman, widowed, without children, with a low level of education, with multimorbidity, and who perceived her health and quality of life positively. Most of the participants were very concerned about COVID-19 and its effects. In fact, 38% had been diagnosed with COVID-19, of whom 20% were admitted to hospital and 20% had suffered negative impacts, such as pain and neurological problems. In addition, 70% of the residents remained confined to their rooms, which increased their perceptions of loneliness and social isolation. The worst-rated aspects of the nursing home resulted from the restrictive measures imposed on nursing homes during the pandemic. This research offers useful material for understanding the pandemic and its consequences from the perspective of the older institutionalised population, which could provide insights for designing public policies.

## 1. Introduction

On 31 December 2019, the first case of SARS-CoV-2 infection was reported in China and on 11 March 2020, taking into account the virus’s spread and the severity of cases, the WHO declared a pandemic. The rapid advance of the disease, coupled with scientific ignorance of the virus, led to a global paradigm shift that drastically and suddenly transformed lifestyles. Although the epidemiological behaviour of the virus was international in scope, its impact differed from one country to another.

Scientific evidence shows that age is a risk factor associated with greater lethality of the virus [1,2]. According to data from the Spanish National Institute of Statistics, 19% of the Spanish population in 2021 was 65 years of age or older; these socio-demographic characteristics facilitated a rapid expansion and profound impact of COVID-19.

The virus did not spread uniformly throughout the territory [3]; some enclaves showed higher morbidity and case fatality rates than others, with nursing homes for the elderly being places of high incidence and mortality [4]. In this regard, the pandemic posed a challenge that made it necessary to take drastic measures to control the virus and prevent it from spreading. A series of regulations and recommendations were issued at different scales, resulting in a complex regulatory network and the overlapping of different territorial decision-making levels, as residential care and long-term care-related competencies are highly decentralised in Spain.

General data indicate that more than half of people who are diagnosed with COVID-19 are over 60 years of age [5], a population segment that most frequently presents more severe symptoms of the disease [6]. According to data from June 2022 from the Institute of Older People and the Social Services (IMSERSO in Spanish) [7], there has been an excess of 33,052 deaths in care homes for the elderly in Spain since the beginning of the health crisis. In absolute terms, Madrid is the region where most institutionalised older people have died from the coronavirus, followed by Catalonia, Castille and Leon and Castille-La Mancha [8]. Likewise, the Spanish nursing home model has a series of specific characteristics that may have contributed to making these centres focal points of contagion and mortality. One of the main characteristics is the high level of ageing of the population. In 2019, the population aged over 80 years represented 79% of all residents [9]. Another factor is the location of many of the residences for older people in the city centre [10], which is denser than other areas. This may have served to increase the spread of the virus as nursing homes could not sectorize modules or isolate outbreaks [11,12].

## 2. Theoretical Framework

The intersection between studies on ageing, residential spaces and COVID-19 has prompted multiple research projects and reflections in the Spanish context, including studies on the virus’s consequences according to nursing home type [5], health recommendations to manage the pandemic [13], the precariousness and fragility of the Spanish residential care system [14,15], transitions to other care models [11], and ageism and discrimination against the older population [16]. There have also been literature reviews on international research linked to the status and progress of COVID-19 in nursing homes [17,18,19]. COVID-19’s impact can be measured in health terms [20] but also in sociological [21,22], psychological [23,24], and demographic terms [25,26]. Coronavirus infection had a direct and very surprising impact on nursing homes. What was understood as a new phenomenon was in fact the consequence of situations inherited from a long-established care system that was not ready to tackle a situation such as the one generated by the pandemic.

This is partly because older adults have been scarcely acknowledged in modern society. They are generally considered a population group of minor social importance and are sometimes affected by ageist stereotypes, prejudice, and discrimination. This has impacted the social-health system in providing adequate treatment for their health, as evidenced during the pandemic [27] when additional resources were needed to meet healthcare demands. There is plenty of evidence that the virus affected the older population group to a greater extent due to multimorbidity conditions and because part of this population lives in nursing homes where the virus was easily spread. This same pattern occurred in a generalised manner in different countries [28,29].

Multimorbidity conditions were compounded by institutional precariousness in caring for patients in nursing homes [30,31], which became places where the virus was easily spread, rather than safe places where it was contained [32]. This precariousness in human and material resources resulted in a confusing and sometimes chaotic situation [33]. The conditions imposed by the long-term care system management and governance model also did not help to control the pandemic [34]. During the pandemic, the need to integrate specific health services with social and healthcare became apparent, and this situation occurred not only in developed countries [35] but also in less advanced ones, where a lack of coordination and diffuse administrative responsibility made it more difficult to manage the pandemic in nursing homes.

In an attempt to control infection in nursing homes, prevention measures were imposed in many of them, including confinement within rooms without the possibility of using common residential spaces. During the first few months of the pandemic, research emerged in an attempt to document the possible consequences of this measure, both in the general population [36] and specifically in the institutionalised older population, in addition to studies focused on the possible risks of cognitive and functional deterioration associated with confinement [37,38]. 

Despite the initial idea that the older population was the most vulnerable and most likely to suffer the emotional consequences of lockdowns, such consequences seem to have had a greater impact and degree of affectation among the younger and female populations [39]. Indeed, “gender (being a woman), living with chronically ill people, living with dependent people and working as a healthcare worker” [39] (p. 39) seem to be four of the socio-demographic variables that function as predictors of increased distress. In addition to these predictors, one could add having previously received mental health treatment [40] and a higher perception of threat linked to low educational levels [41]. The data suggest that the older non-institutionalised population “does not show negative scores in psychosocial and health aspects during the pandemic, despite being a group at risk for the development of distress during this time” [42] (p. 157). In short, the a priori assumptions about the pandemic’s effects on older people do not match the conclusions that have been reached over time, which in fact relate the younger population to higher levels of vulnerability [43,44,45].

Some studies suggest that the health conditions of older adults in nursing homes have worsened as a result of being isolated and, in many cases, having suffered from the disease [6]. The clinical effects derived from being affected are weight loss [46], depressive symptoms, deterioration of cognitive functions [47], increased frailty as a consequence of a lack of mobility [48] and insomnia [49]. In addition, the pandemic forced nursing homes to lock down, preventing visits by family and friends. A lack of human contact, especially the disappearance of usual close social ties, can trigger physical, psychological and cognitive consequences [50]. From the very outset of the pandemic, maintaining ties with relatives was considered a measure of relief against the disease’s consequences [51]. Simultaneously, the lockdown and social distancing measures led to professionals feeling emotional exhaustion due to difficult working conditions; residents suffering isolation, loneliness and a change in daily routines [4]; and relatives experiencing restlessness and anguish.

The pandemic has also highlighted certain pre-existing problems in the older institutionalised population, such as loneliness [52,53,54]. In many cases, the compulsory physical and social distancing led to situations of total isolation [17,33,55,56], with the consequences that this entails both for physical health [57], including mortality [58], and for psychosocial capacities [59]. In this regard, isolation in Spanish nursing homes during certain stages of the pandemic became drastic, with residents being confined to their rooms for days or weeks at a time. 

Given the radical changes older persons were subjected to in residential environments, the question arises as to how the pandemic affected residents’ quality of life. Results in this respect are by no means homogeneous, and a study conducted in Belgian nursing homes observed that the pandemic, and being confined in particular, triggered a loss of quality of and meaning in life for residents by depriving them “of freedom, social life, activities and autonomy” [60] (p. 14); this is in line with the results of similar studies conducted in other contexts, according to Sayin Kasar and Karamar [61]. However, other studies have shown that older people “rated their quality of life, life satisfaction and well-being during the pandemic better than younger people” [62] (p. 1).

Methods of coping with the pandemic are linked to value systems and collective imaginaries that are rooted in certain ages. Thus, narratives articulated about older people’s vulnerability, frailty and helplessness could have been reinforced during the pandemic by COVID-19 [14,63], underscoring the spectre of ageism. In addition, the media may have contributed by portraying older people negatively or unfavourably [64]. Ageism has a netlike shape that spreads and permeates social issues very subtly, causing individuals, social agents and institutions to reproduce discriminatory logic. Different studies, such as the one by García Soler et al. [65], highlight intergenerational tensions that escalated during the pandemic and focus on the violation of certain basic rights as a consequence of sustaining harmful, socially shared representations and imaginaries. 

These were the premises upon which research on homes for the elderly in the Region of Madrid was carried out between June and October 2021 through a survey divided into several thematic modules to be answered by residents. The research aimed to answer the question of how the pandemic affected the population in Madrid nursing homes. The impact of the pandemic is understood in a complex and integrated way in relation to various dimensions such as physical health, mental health, emotions about the pandemic, quality of life, the residential environment and the self-perception of ageing. This article discusses the planning, study design, methodology and logistics of the fieldwork and provides an overview of the results and characteristics of the population studied.

## 3. Materials and Methods

The Project “Nursing homes and COVID-19. Environments of older people as protectors in health emergency situations” has used a methodology that combines quantitative and qualitative information. To gather quantitative data, a questionnaire was used to measure, quantify and systematise the experience of and effects on older persons living in nursing homes during the COVID-19 pandemic. The qualitative perspective, which will be developed later, will allow us to delve into meanings, motivations and experiences through a discursive and narrative analysis of social agents (residents, workers, family members and institutions).

### 3.1. Technical Specifications

The first stage involved conducting a sampling process in nursing homes and, in the second stage, older people were selected to participate. The sample stratification criteria were the nursing home’s ownership (public or private), its geographical location (urban or rural) and its size according to the number of places for residents. A total of 42 nursing homes (Table 1) and 447 residents participated in the study, with a similar distribution according to nursing home ownership and location.

The initial sample size (600 surveys) was calculated with a maximum sampling error of ±5% for estimating percentages under the hypothesis of maximum variability (p = q = 0.5) and at a confidence level of 95%. The calculation assumed a design effect of 1.5 (the ratio between the variance of the estimator under the type of sampling used and the variance of simple random sampling). The sample had to be redesigned due to difficulties in completing the fieldwork (see Section 3.2), maintaining the initial dispersion and the ratio between first (nursing homes) and second stage (residents) units. This significantly reduced the design effect, which, in this case, could be approximated by 1.2. Thus, the maximum absolute sampling error could be bounded by ±4.8%, keeping the other sampling parameters stable.

The nursing homes were selected in terms of probability proportional to their size (number of residential places), while participants were selected by simple random sampling (approximately 10 persons from each nursing home). Other inclusion criteria according to the information provided by the nursing home were male/female, being 60 years of age or older, not suffering cognitive impairment and having given consent to participate in the research. The fieldwork was submitted to the Bioethics Committee of the Spanish National Research Council (CSIC), which gave its approval in June 2020.

Accessing the nursing homes involved great effort from the start of the fieldwork due to access restrictions put in place as the pandemic evolved during 2020 and the first half of 2021. Indeed, 95 nursing homes were contacted and details of only 42 homes were obtained. Data were collected at different paces as the number of contacts required to obtain data was lower in public nursing homes (2.1 contacts on average) than in private ones (Section 5.4). Similarly, the average number of response days between first contact and completion of the survey was much shorter for public (12.7 days) than for private (24) homes. In addition to pandemic-related reasons, responses were delayed by difficulties in contacting the person responsible, the time available to answer the call, the need for the management team to reach a consensus about giving their consent to participate and obtaining the permission of the nursing home’s parent company, among others.

The nursing homes that decided not to participate were replaced by another 13 reserve nursing homes until the final number was reached. This process of contacting the nursing home/affirmative response/negative response and substitution, if applicable, took a relatively long time—more than two weeks for the private nursing homes and more than 10 days for the public nursing homes. Another factor was the need to make a high number of contacts (between three and four). Different reasons were given for not participating, ranging from not having enough older people without cognitive impairment, no specific reason, difficulties in organising the survey while the nursing home staff were working, managers having little time available, unexplained delays in the responses, etc., which might suggest that staff were overloaded with daily chores or a possible aversion to participating, given the difficult circumstances in which the research was carried out.

A pilot sample was carried out in early May 2021 in two public nursing homes to test the feasibility of the questionnaire and other aspects to be considered in the fieldwork, and this led to two questionnaire questions being changed. The fieldwork was carried out between June and October 2021 by an opinions research company whose pre-trained interviewers had the questionnaire programmed into a computer application. The personal interviews lasted approximately 25 min on average. Once the information was collected, it was sent for review by a market research company manager before being added to the database.

The interviewers went to the nursing homes properly identified. For this purpose, several project information documents were prepared: (1) approval of the Spanish Scientific Research Council Bioethics Committee (Ref. Nº. 114/2020), (2) official resolution adopted for the project by the Spanish Scientific Research Council (Ref. Nº. 202010E158), (3) information about the project’s objectives and (4) a general information sheet for participants (participation commitments, research team’s duties towards each participant, informed consent).

### 3.2. The Questionnaire

The questionnaire contained topics of interest for the project, which were drawn up and designed by a multidisciplinary expert committee. Finally, 20 questions of different types (simple, compound, multiple, multi-response, Likert-type and rating scales) were presented thematically: personal circumstances during the pandemic, health, feelings and methods of coping, residential environment and activities, quality of life, ageism, and attitude towards own ageing (Appendix A). Additionally, basic and objective socio-demographic information was obtained (sex, age, education level, marital status and number of children), together with objective clinical data about the participants, which were drawn from medical reports prepared by health personnel and made available to the project by each institution, with the sole exception of one nursing home (eight participants) that did not grant access to health data.

#### 3.2.1. Personal Circumstances during the Pandemic

This block included questions that sought to paint a general overview of how the person had been affected by the pandemic (Appendix A, Personal situation). The first question asked about whether or not the person had been affected by the disease and the impact of possible effects in the case of having been infected. A subjective assessment was also obtained that was different from the question on the diagnosis of the disease (Appendix A, Health). Then, they were asked about their opinion and level of concern regarding the pandemic and whether their satisfaction with life had been affected. Another material used was the shorter version of the COVID-19 fear scale, FCV-19S [66], an instrument designed specifically for the pandemic, which has been widely used [67] and validated in different contexts and countries [68], including Spain, although mostly with the general population [69] and to a lesser extent the older population in nursing homes [70]. Finally, we asked questions about a series of protective measures that were implemented due to the norms established by the health administration [71,72].

#### 3.2.2. Health

Understanding residents’ health was one of the most important objectives of the research (Appendix A, Health). Perceived health status is simple to assess and this approach is widely used and validated in Spain, based on the 2017 National Health Survey and 2020 European Health Survey in Spain by the National Institute of Statistics (INE). Respondents were also asked about the diagnosis of COVID-19 and its subsequent development. These questions were obtained from the Questionnaire for Assessing the Impact of the COVID-19 Pandemic on Older Adults (QAICPOA) [73] which examined the contagion, its diagnosis and hospitalisation, the number of days of treatment, and the care received by professionals. The disease’s mental impact was explored through two measures that were easy to apply and understand. The first measure was the Abbreviated Geriatric Depression Scale (GDS) [74], adapted to Spanish in a simplified 5-item version [75,76,77] which, although it is applied to older persons, has not been used in nursing homes. 

Perception of loneliness was measured by a unique question [78], adapted for the older population [79]. This question was consistent with the results obtained with other abbreviated scales, such as the De Jong Gierveld 6-Item Scale [80,81] and the UCLA 3-Item Loneliness Scale [82], which measure the perception of loneliness and its intensity.

Health-related information was obtained with questions about satisfaction with the care provided by staff (medical, nursing, physiotherapy and auxiliary) and services in the nursing home (technical resources).

#### 3.2.3. Feelings and Coping 

This set of questions responded to interest in understanding the emotional mechanisms and strategies for coping with COVID-19 (Appendix A, Feelings and coping) and involved using the 10-item Positive and Negative Affects Schedule (PANAS) [83], which has been validated in multiple contexts and population groups [84]. In Spain, it has been widely used among the general population but has had limited use among older people [85].

Resilient coping explores positive adaptation to stressful or traumatic experiences through a brief scale (Brief Resilient Coping Scale, BRCS), which was constructed from the response to items with a single positive valence [86]. It has been used on a wide scale [87], even in older people. This scale was adapted by Tomás et al. [88] and Navarro-Pardo et al. [89] for the Spanish context.

#### 3.2.4. Residential Assessment and Activities Performed

As this study examined older people living in nursing homes, their opinions on such settings during the health crisis were considered relevant on account of the difference with the family homes and because of the high incidence of the pandemic in nursing homes [4]. 

The residential model in Spain, which is linked to stay and care costs, offers two possibilities: have one’s own room/shared with a partner/other family member or share a room with another resident. This was explored with a simple question to relate it to the advisability of staying in the room in order to avoid catching the virus. In addition, this restriction regarding the use of nursing homes’ common areas is linked to possible limitations in carrying out activities, whether these are “survival” activities (eating or walking) or leisure time ones.

The specific scale on the Assessment of Aspects of the Residential Environment (EVAER, of the Spanish acronym) measures various aspects of daily life in nursing homes by means of a 5-level Likert scale (very good, good, fair, bad, very bad). The Factor Analysis (by principal components and Varimax rotation) showed high communalities and identified four factors (72.2% of the accumulated variance, with an adequate KMO measure (Kaiser-Olkin-Mayer test, 0.823), which were interpersonal relationships in the nursing home, aspects of the nursing home (resources, security and management), mobility (internal and external) and intimate space (having a room available). With this information, four subscales were constructed (relationships, nursing home aspects, mobility and intimate space) by grouping the scores of items for each one. So, higher values indicated a better assessment. Application of Cronbach’s Alpha coefficient showed values above 0.7 as a threshold for accepting the hypothesis of reliability of the internal consistency of the items with the subscales [90]. 

Engaging in activities is an essential aspect to give meaning to daily life in a nursing home as a social space for the older population. In this research, we designed questions about eight activities drawn and adapted from previous studies [91,92,93] and other bibliographic references [94,95,96], but which did not initially follow any particular classification scheme. Respondents were asked for information on how often (never = 0 days, yearly = 3 days, monthly = 12 days, weekly = 52 days, daily = 365 days) they engaged in up to eight types of activities and any changes during the pandemic (Appendix A, Residential assessment). Using k-means Cluster Analysis and taking the frequency of activities as input variables, four groups were identified: people who use electronic devices, people who engage in volunteer activities/doing things for others, inactive people and people who engage in religious practices. In addition, respondents were asked their opinion on whether the pandemic had affected their performance of activities compared to their pre-pandemic situation.

#### 3.2.5. Quality of Life

Quality of life is a scientific construct with a strong social value. It is used in a large number of studies and disciplines for various purposes and is a term that is used in everyday life. However, besides its usefulness, it is characterised by difficulties associated with its definition and measurement, precisely because of the diversity of possible applications. This study included a measure oriented by its objectives (Appendix A, Quality of life), although without previous treatment in the literature on COVID-19, namely, the FUMAT-24 scale [97]. This consists of 24 items grouped into eight subscales and covers essential aspects to measure the quality of life of older people in nursing homes. 

#### 3.2.6. Ageism and the Self-Perception of Ageing

The term ageism [98] refers to the creation, existence or maintenance of stereotypical images based on prejudices about ageing and old age, which could promote age discrimination. This can lead to older people being marginalised and segregated by society but also to them assuming their own subordination by internalising these stereotyped and preconfigured ideas. This marginalisation is now more apparent due to the lack of sufficient attention paid to the potential impact of COVID-19 in nursing homes [30]. Ageism has also become more evident during the pandemic [99], as attested by the World Health Organization itself, which went so far as to launch a campaign to combat it [100].

The questionnaire included six questions on the three dimensions of ageism (prejudice, stereotypes and discrimination), adapted to the situation caused by COVID-19 and placing the older person’s perspective at the centre (Appendix A, Ageism).

Likewise, the older residents’ perceptions of their own ageing were assessed, as recommended by Levy et al. [101], through the Five-item Attitude Toward Own Aging subscale (ATOA) of the Philadelphia Geriatric Center Morale Scale (PGCMS) [102,103].

#### 3.2.7. Socio-Demographic Characteristics 

This topic was covered by the basic objective variables (sex, age, marital status, education level and number of children) obtained from the documentation provided by the selected nursing homes and taken in parallel with the questionnaire (Appendix A).

## 4. Data Management and Statistical Analysis

After the fieldwork, the data file was reviewed exhaustively using the SPSS-v28 pro-gramme for debugging purposes. The first step consisted of ensuring that the information in the database was consistent with the questionnaire and redefining the properties of the variables in terms of levels of measurement (nominal, ordinal, scale). In addition, a mnemonic name was assigned to the variables to allow their immediate identification at any time during the study, and the categories of the nominal and ordinal variables, their numerical codes and literal labels were revised. An essential part of this standardisation process consisted of examining the missing answers to some questions (“Don’t know/No answer”); in filtered ones these answers were not regarded as missing. In this same regard, we checked whether the criterion of inclusion of participants by age from 60 years onwards had been met. Furthermore, new variables were generated by recoding the categories of some questions (nominal, ordinal and measurement scales) and counting others, such as, for example, the effects derived from having had COVID-19 or the diseases and medications listed in the direct information provided by each nursing home on the participants. 

Specific mention should be made of the scale analysis. Taking the scientific literature as a reference, various statistical techniques were used, such as Factor Analysis by principal components and the internal reliability test of the items—applying Cronbach’s Alpha or Kuder-Richardson KR-20—to determine some basic psychometric characteristics. This procedure was performed with the FCV-19S, PANAS, BRCS, EVAER, Activities, FU-MAT-24, Ageism and Self-perception of ageing scales.

To obtain the basic results presented in this study, frequencies were calculated to check the consistency of the information and draw initial ideas about their distribution. As expected, they were not distributed evenly, and this may have been related to the respondents’ traits, which is why statistical relationships between variables had to be obtained by means of contingency tables and measures of association. Thus, use will be made of univariate and bivariate analysis techniques and other more complex, multivariate and statistical modelling techniques adapted to each specific data. The internal consistency of the ATOA subscale was run using the Kuder-Richardson KR20 reliability coefficient (for dichotomous items), following the Stata module by Caci [104] and adapted to Stata v15.5 for this study.

## 5. Results

The sample consisted of 447 participants, representative of institutionalised older people in the Region of Madrid (Spain). In relation to the socio-demographic characteristics (see Appendix A), the population had a mean age of 83.8 years, and 63.1% were women. Almost half of the sample (48%) had no living children and 50% of the sample were widowed. Thus, the most common profile of the sample was that of an older woman without children, widowed and with a low level of education. 

### 5.1. How Did Respondents Rate Their Situation during the Pandemic?

Slightly more than a third of the respondents reported having had the disease, but without effects, given the living conditions in the nursing homes, while 55% stated that they had not contracted it. Only one in 10 had experienced after-effects (sequelae), the most frequent symptoms being pain and neurological problems. However, just over a third (38%) were diagnosed with COVID-19 by a medical professional and, of these, two out of ten stated that they had been admitted to hospital for this reason (Appendix A). 

Faced with the health crisis generated by the pandemic, concern about COVID-19 among older residents was widespread (83%), and in fact, six out of ten were very worried. Concomitantly, fear of COVID-19 was not, in general, a very present feeling (score of 18, between 7 and 35), although some of its components reached a relevant proportion, such as fear of death and of the SARS-CoV-2 virus itself. In contrast, more than half of respondents (53%) reported feeling as satisfied with their lives as before, compared to 43% who reported being less satisfied.

### 5.2. How Did the Pandemic Affect Their Health and How Did They Manage It?

The participants’ medical reports showed that six out of ten people had more than five diseases, with heart (83%) and bone (63%) diseases being the most common (Appendix A). Perceived health was good or very good for almost 60% compared to 13.6% who considered it to be poor. Data on medical care revealed that 35% of the residents had been seen by a physician or specialist during the pandemic, while 16% had had an appointment cancellation. Satisfaction with the care received from healthcare personnel and material resources was very high (more than 80%).

Depression reached a mean score of 1.7 (minimum: 0, maximum: 5) on the GDS-5 scale. More than five out of ten older people had no symptoms (values 0 and 1), the rest were suspected of having depression, with almost 20% scoring at a level indicating moderate or severe degrees of it.

### 5.3. Did COVID-19 Affect the Way They Feel and Behave?

It was confirmed that positive effects prevailed over negative ones; thus, the balance between the two was positive. However, their scores (11.4 and 7.8, respectively) were below the expected mean for the scale (12). Feeling active and attentive were the most frequently reported positive emotions (Appendix A). 

In the nursing homes, 47% of people reported never or almost never feeling lonely. Among those who always or almost always felt lonely (13%), half communicated feeling lonelier than before the pandemic.

More than 50% of respondents reported being fairly or very prepared for the COVID-19 situation and 40% believed they could take something positive from it, while 15% to 20% related feeling bad about themselves or being caught up in negative thoughts as a result of the pandemic. More than 60% of the residents agreed with developing attitudes of control regarding the pandemic situation.

### 5.4. Is the Nursing Home an Adequate Living Space?

Most residents had a single room or shared a room with a spouse/partner/family member, while less than three out of ten shared a room with another resident. A high percentage of people (70%) stated that they were always in the room during the COVID-19 pandemic. Despite the movement limitation measures, the rest enjoyed some mobility outside the room, with 26% making limited use of the residential spaces, mainly for eating and walking in common areas and, to a lesser extent, for social gatherings (Appendix A).

In general, the residential assessment was good in all the domains analysed, especially regarding safety and relationships with family members, in contrast to mobility in and out of the nursing home, which was less valued due to restrictions during the pandemic. A more detailed analysis of the residential assessment identified four subscales linked to people’s behaviours (their relationships and mobility) and other aspects related to living in a nursing home, such as their safety, privacy or how their way of life was managed. As a complement, another residential satisfaction-associated question sought to compare the situation during the pandemic with that prior to the spread of the disease. The results indicated that the pandemic had not altered their assessment of their stay, since six out of ten stated that it was the same and a quarter stated that it was better.

On a daily basis, residents in the nursing home (more than 80%) engaged in simple activities (listening to the radio, watching television) or activities of little physical relevance (gentle exercise or walking). Activities that required a certain degree of personal motivation (reading books, using electronic devices, doing crossword puzzles and playing board games) were practised daily by less than 50%, as were doing handicrafts, going to chapel or doing activities to support other people. The four clusters of people who engaged in activities responded to a bipolar profile in which many older people with low levels of activity, including those who engaged in religious practices, contrasted with somewhat smaller groups of people with more active behaviours who were interested in the use of electronic devices or who did activities for the benefit of others or volunteering. 

How did these habits change during the pandemic? Between 70% and 80% of the surveyed population maintained their activities during the pandemic (Appendix A). The increase in doing activities was barely noticeable in most of them, and they reduced the frequency of walking and/or exercising as a consequence of mobility restrictions (26%), doing handicrafts or going to chapel (20%), doing support activities (12%) or doing things for others (12%).

### 5.5. How Has Your Quality of Life Been Affected during the Pandemic?

In general, all domains of quality of life scored highly, with an overall average of almost 10 out of 12 (Appendix A). The domains with the highest values were relationships within the nursing home with all people (residents, caregivers, others), respect for a person’s rights, material wellbeing (comfort, absence of barriers, availability of material things) and personal development related to personal autonomy. The rest scored below average, including the domain measuring social inclusion (participation in activities and difficulty in finding support) and emotional and physical wellbeing derived from living in a nursing home, which is linked to satisfaction with current life and oneself, mobility problems and incontinence. The lowest value, which was approximately 7.5, was the capacity for self-determination, which is related to the difficulties that people in nursing homes experience in making plans for the future, choosing how to spend their free time or preventing other people from making decisions for them. Consequently, and as an aggregate measure, overall quality of life had a high value, with more than half of the residents above the average (79.6 out of 95).

### 5.6. Are the Older Adults in Nursing Homes Ageist and How Do They Perceive Their Own Ageing?

Older adults in nursing homes are convinced that they are more responsible and competent than other population groups in following measures to cope with the virus (78%), making their own decisions freely (61%) and, in the event of a shortage of health care resources, obtaining the same access as other population groups (86%) (Appendix A). Regarding the less positive dimension of ageism, 41% confirmed that they agreed that older people are a burden on health and social services. Eight out of ten considered that this group suffered more emotional impacts than other age groups with respect to the news about the consequences of the coronavirus and almost 50% stated that, considering the risk of contagion, meeting with older people is more dangerous than meeting with other younger population groups. The scale of self-perception of ageing revealed relatively negative scores, with a mean value of 1.8 (minimum: 0, maximum: 5). Among the aspects consulted, approximately half of the people interviewed stated that they felt less useful as they aged and approximately 38% confirmed that things seemed worse to them as they got older.

## 6. Conclusions

Since the outbreak of the pandemic, many social behaviours have changed in addition to many research procedures in terms of design, data management, analysis and diagnosis of situations. One indicator of this new reality is the exponential increase in studies assessing the multiple aspects and effects of COVID-19. In essence, urgent research was launched and processes and circumstances were diagnosed, and unconsolidated evidence and a diversity of scientific products of unequal value were published [105,106,107].

Older adults have been the subject of research due to the extraordinary and severe incidence of the pandemic among them. Multiple studies have approached this group from different perspectives—mostly considering the older population as part of the general population—by means of online surveys (on the Internet), which limits access to those who do not have the capacity to respond (digital divide). In addition, few studies have focused on nursing homes for older persons and, when they have done so, the topics of interest were linked to the study of residential management during the pandemic, the design and application of regulations, or the impact on workers. However, residents’ previous situations and habits (health, activities, behaviours, etc.) and the possible changes that occurred due to the pandemic were not adequately examined in these studies.

One of this project’s contributions is to have chosen older people living in nursing homes as the subject of study as this is not frequent and signals a methodological shift that enables direct access to this population’s opinions, perceptions and circumstances. In addition to its scientific interest and value, this calls for demanding fieldwork that sometimes involves limited access, the execution of which is conditioned by the bringing together and understanding of various agents, disciplines and interests. However, diagnosing the situation of older aged in nursing homes based on a consensus between scientific analysis and residential environment managers is essential to improving older people’s quality of life so that they can live their last stage of life in conditions of dignity and respect for their rights as persons. 

Researching institutionalised older adults and fairly inaccessible residential environments is a task that is not likely to be free of difficulties, thereby having a relative impact on the results obtained. Given the urgency of the decision-making process, selecting nursing homes in the Madrid Region is a limitation of this study. This was driven by the need to carry out fieldwork that was as controlled as possible during the pandemic. We are thus forced to assume that the situation in this geographical area can be extrapolated to the rest of Spain, which may not be true. However, the selection of nursing homes was conditioned by their distribution in terms of owners and managers, and this is true in Madrid, but may not be so in other Spanish regions. Some studies have revealed serious deficiencies in the Spanish nursing home system, such as a lack of material resources (poor location, inadequate space, lack of equipment, excessive use of restraints and medications) and human resources (lack of personnel, labour insecurity). This is aggravated by the lack of inspections and the low level of fines for infractions [7,38]. This diagnosis suggests that a modification of the nursing home care system by the Spanish government is required.

In addition, and in accordance with the study’s purposes, the data collection had to be directly from the respondent, which necessarily involved selecting responsive residents, i.e., people without cognitive impairment or with only mild impairment, according to the information provided by the nursing homes. This decision left out all those people with worse mental health conditions. Therefore, the information obtained must be read with a qualified approach, taking into account that part of the resident population was excluded from participating. The experience both of relatives and of the nursing homes’ workers and managers was discarded in this quantitative phase to be explored qualitatively a posteriori.

Perhaps this too is a limitation, but the fact that there were no previous or parallel studies to help the research team select the right content led us to identify the research topics in terms of social and health dimensions, in line with the project’s objectives. The result has certainly been satisfactory because the response rate was good, the quality of the data was high and the analyses carried out so far are highly consistent, reinforcing the interpretation and diagnosis of these situations. The questionnaire’s results will be contrasted with the opinions and experiences of residents, family members and workers in later phases.

In short, and despite its limitations, this research, which focused on older nursing home residents and was designed with the urgency and need to assess their opinions and experiences during the pandemic will pave the way to reflect on decision-making concerning older people’s lifestyles during health crisis situations.

## Figures and Tables

**Table 1 ijerph-19-16629-t001:** Distribution of cases according to ownership and location.

Location						
Ownership	Nursing Homes			Residents		
	Municipality of Madrid	Madrid Region *	Total	Municipality of Madrid	Madrid Region *	Total
Private	12	10	22	112	108	220
Public	11	9	20	133	94	227
Total	23	19	42	245	202	447

* Madrid Region excluded the municipality of Madrid.

## Data Availability

Not applicable.

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
