# Peer review of "The Impact of COVID-19 on Nursing Homes: Study Design and Population Description"

_ijerph, 2022, doi:10.3390/ijerph192416629_

Round 1

Reviewer 1 Report

1.     The authors should analyze the difference of samples’ perceptions between 220 in private nursing homes and 227 in public nursing homes.

2.     The questionnaire’s validity and reliability should be described in the text; not only describing in the Table (supplementary).

3.     The strategies to protect the residents living in nursing should be discussed by the results of statistical analysis. Also, to explore the protective strategies comparing to government protect strategies seems important.

Author Response

REVIEWER 1

The authors are very grateful for the support provided by the reviewer in his/her task. Please find below the authors’ answers

Reviewer question

  1. The authors should analyse the difference of samples’ perceptions between 220 in private nursing homes and 227 in public nursing homes.

Authors’ answers

For a better understanding the text in lines 189-191…:

The aim behind this article is the planning, study design, methodology and logistics of the fieldwork, as well as providing with an overview of the results and characteristics of the population studied.

…must be modified

The aim behind this article is the planning, study design, methodology and logistics of the fieldwork, as well as providing descriptive results about the characteristics of the population studied. It does not intend, therefore, to conduct a statistical analysis of the data nor an intervention project, as is usual in scientific literature on project design studies. A more in-depth analysis is being channelled into several articles, some under review, and others already under development.

Reviewer question

  1. The questionnaire’s validity and reliability should be described in the text; not only describing in the Table (supplementary).

Authors’ answers

The text in lines 269-270:

The questionnaire contained topics of interest for the project, drawn up and designed by a multidisciplinary expert committee.

…has been amended as follows:

In order to ensure the validity of the questionnaire content, it was developed by a committee of experts, which built it up through in-depth bibliographic reviews and ad-hoc discussions. The validity and reliability of measures deployed are being worked upon in specific papers. Finally, there were 20 questions (…)

Reviewer question

  1. The strategies to protect the residents living in nursing should be discussed by the results of statistical analysis. Also, to explore the protective strategies comparing to government protect strategies seems important.

Authors’ answers

The paper’s objectives do not include either statistical analysis or further discussions, just a project's design and implementation

Reviewer 2 Report

The article could be publishable, but it would be necessary to take into account certain aspects for its improvement:

· In my opinion, the information on the evaluation instruments is not sufficiently detailed, specifically in those variables that are not measured by means of standardized instruments.

· There is talk about "elderly home project and covid-19", but in reality an intervention or prevention project is not presented, but a descriptive study on certain variables

· I think it would be convenient to review how the objectives of the work are stated

· In line 219, sex is indicated as an inclusion criterion. What do the authors refer to?

· The statistical analysis is quite simple, limiting itself to describing the results of the variables but without establishing any type of analysis related to the relationship between them. This could have been useful, for example, to find out the relationship between certain variables and the quality of life of users.

Despite the fact that it is stated as an objective, in the conclusions, no emphasis is placed on aspects that it would be convenient to work on in the elderly as a consequence of the health crisis, or what aspects should be taken into account for future action programs in the face of new situations of emergency

Author Response

REVIEWER 2

The authors are very grateful for the support provided by the reviewer in his/her task. Following the authors respond to the reviewer’ comments

Reviewer question

In my opinion, the information on the evaluation instruments is not sufficiently detailed, specifically in those variables that are not measured by means of standardized instruments.

Authors’ answers

All measures employed are displayed in Table S.1. Measures included in the COVID-19 Residential Homes Survey (Madrid Region, Spain). Its title and other small contents have been changed to gain clarification. This table covers the questions’ items, minimum and maximum values taken and Cronbach alpha, if applicable. Section 3.2. in the manuscript also summarizes these measures along with the references. Table S.1 has been resent to the editor

Reviewer question

There is talk about "elderly home project and covid-19", but in reality an intervention or prevention project is not presented, but a descriptive study on certain variables

Authors’ answers

As stated above, the paper does not analyse an intervention initiative but a research project. We refer to "Project" as "Study" in the text in most cases. So, for further clarification and when appropriate, we have changed the word "project" in lines

193 with 'research project',

259: delete ‘project information’

422: delete the word ‘project’

263: maintain 'project',

in lines 265, 277, 420, 553 and 591 with 'study'

Reviewer question

I think it would be convenient to review how the objectives of the work are stated.

Authors’ answers

Text rewritten in lines 182-186: The research aimed to answer the question of how the pandemic affected the population in Madrid nursing homes, to understand the consequences of COVID-19, as well as the particularities of this impact on the ways of living and experiencing such a situation as the pandemic. This impact..

Reviewer question

In line 219, sex is indicated as an inclusion criterion. What do the authors refer to?

Authors’ answers

Text rewritten in Lines 223-227: The residences were selected by probability, proportional to their size (measured in number of places). Then, in each residence, the people to be interviewed, approximately ten, were selected by simple random sampling, and distributed by age and sex. Inclusion criteria to participate included being 60 years of age or older, not suffering from cognitive impairment and having consented to participating in the research. The fieldwork was submitted (...)

Reviewer question

The statistical analysis is quite simple, limiting itself to describing the results of the variables but without establishing any type of analysis related to the relationship between them. This could have been useful, for example, to find out the relationship between certain variables and the quality of life of users.

Authors’ answers

The paper’s objectives do not include either statistical analysis or further discussions, just a project's design and implementation

Reviewer question

Despite the fact that it is stated as an objective, in the conclusions, no emphasis is placed on aspects that it would be convenient to work on in the elderly as a consequence of the health crisis, or what aspects should be taken into account for future action programs in the face of new situations of emergency

Authors’ answers

For greater clarification, the authors would like state that the research team is now carrying out more specific papers on the assessment of residential features, on the fear of COVID-19 scale validation, on the health outcomes linked to loneliness and depression, on the activities performed by residents and on the self-perception of ageing and ageism. Different analytical approaches are being applied according to the papers' objectives.

Round 2

Reviewer 1 Report

This paper could be published in this journal.